

# Marine biodiversity research in the Ryukyu Islands, Japan: current status and trends

James Davis Reimer[1,2,*], Piera Biondi[1,*], Yee Wah Lau[1,*], Giovanni Diego Masucci[1,*], Xuan Hoa Nguyen[1,3,*], Maria E.A. Santos[1,*] and Hin Boo Wee[1,*]

[1] Graduate School of Engineering and Science, University of the Ryukyus, Nishihara, Okinawa, Japan
[2] Tropical Biosphere Research Center, University of the Ryukyus, Nishihara, Okinawa, Japan
[3] Faculty of Environment, Vietnam National University of Agriculture, Hanoi, Vietnam
[*] These authors contributed equally to this work.

## ABSTRACT

Marine biodiversity and derived ecosystem services are critical to the healthy functioning of marine ecosystems, and to human economic and societal well-being. Thus, an understanding of marine biodiversity in different ecosystems is necessary for their conservation and management. Coral reefs in particular are noted for their high levels of biodiversity, and among the world's coral reefs, the subtropical Ryukyu Islands (RYS; also known as the Nansei Islands) in Japan have been shown to harbor very high levels of marine biodiversity. This study provides an overview of the state of marine biodiversity research in the RYS. First, we examined the amount of English language scientific literature in the Web of Science (WoS; 1995–2017) on six selected representative taxa spanning protists to vertebrates across six geographic sub-regions in the RYS. Our results show clear taxonomic and sub-region bias, with research on Pisces, Cnidaria, and Crustacea to be much more common than on Dinoflagellata, Echinodermata, and Mollusca. Such research was more commonly conducted in sub-regions with larger human populations (Okinawa, Yaeyama). Additional analyses with the Ocean Biogeographic Information System (OBIS) records show that within sub-regions, records are concentrated in areas directly around marine research stations and institutes (if present), further showing geographical bias within sub-regions. While not surprising, the results indicate a need to address 'understudied' taxa in 'understudied sub-regions' (Tokara, Miyako, Yakutane, Amami Oshima), particularly sub-regions away from marine research stations. Second, we compared the numbers of English language scientific papers on eight ecological topics for the RYS with numbers from selected major coral reef regions of the world; the Caribbean (CAR), Great Barrier Reef (GBR), and the Red Sea (RES). As expected, the numbers for all topics in the RYS were well below numbers from all other regions, yet within this disparity, research in the RYS on 'marine protected areas' and 'herbivory' was an order of magnitude lower than numbers in other regions. Additionally, while manuscript numbers on the RYS have increased from 1995 to 2016, the rate of increase (4.0 times) was seen to be lower than those in the CAR, RES, and GBR (4.6–8.4 times). Coral reefs in the RYS feature high levels of both endemism and anthropogenic threats, and subsequently they contain a concentration of some of the world's most critically endangered marine species. To

Corresponding author
James Davis Reimer, jreimer@sci.u-ryukyu.ac.jp

protect these threatened species and coral reef ecosystems, more data are needed to fill the research gaps identified in this study.

# INTRODUCTION

Biodiversity research provides the basis to guide ecosystem management, and consequently, to preserve services and goods that are critical to the economic value of the planet (*Costanza et al., 1997*; *Mace, Norris & Fitter, 2012*). Moreover, knowledge of biodiversity patterns enables the prediction of possible outcomes from ongoing environmental changes (*Bellard et al., 2012*) and species extinctions (*Chapin et al., 2000*; *Dunne, Williams & Martinez, 2002*). Analyses of species diversity and distribution also allow the determination of biodiversity hotspots. For example, the 'Coral Triangle' hotspot, located in central Indo-Pacific waters, is considered to be the coral reef area with the highest numbers of marine species in the world (*Hughes, Bellwood & Connolly, 2002*; *Toonen et al., 2016*). Nevertheless, there is still a lack of diversity information for most marine taxa (*Appeltans et al., 2012*; *Troudet et al., 2017*), and this problem is especially prevalent in understudied localities including many in the Indo-Pacific. Such data gaps lead to incomplete or inaccurate knowledge of biodiversity patterns, limiting our ability to determine appropriate conservation measures for species and ecosystem functions (*Cardinale et al., 2012*; *Costello, May & Stork, 2013*; *Duffy, Godwin & Cardinale, 2017*).

The Ryukyu Islands (RYS; also known as the Nansei Islands) comprise the southernmost region of Japan and border the northern edge of the Coral Triangle, spanning 1,200 km from Yakushima and Tanegashima Islands (Yakutane sub-region) in the north, across the Tokara, Amami, Okinawa, Miyako sub-regions to the Yaeyama Islands in the south (Fig. 1, also *Nishihira, 2004*; *Fujita et al., 2015*). The RYS includes islands of different geological formation, ages, and sizes (*Kizaki, 1986*; Table 1). The waters of the RYS are all influenced by the warm Kuroshio Current that flows northwards along the west side of the island chain (*Andres et al., 2008*). Thus, the RYS are a marine region of exceptionally high diversity and endemism (*Hughes, Bellwood & Connolly, 2002*; *Cowman et al., 2017*). Moreover, southern Japan and Taiwan rank first in global marine conservation priority when considering high levels of multi-taxon endemism, their high risk of biodiversity loss due to overexploitation and coastal development, and thus need rapid conservation action (*Roberts, Mittermeier & Schueler, 2002*). More than one decade after this initial work, and despite some conservation successes (e.g., *Okubo & Onuma, 2015*; establishment of Keramas National Park in 2016), the RYS are still threatened by rapidly increasing tourism pressure (*Dal Kee, 2015*; *Hirano & Kakutani, 2015*; *Tada, 2015*; *Toyoshima & Nadaoka, 2015*; *Okinawa Prefectural Government, 2016*) and continuous ongoing coastal developments (*Veron, 1992*; *Fujii, Kubota & Enoki, 2009*; *Reimer et al., 2015*). In fact, numbers of tourists visiting Okinawa exceeded those of Hawai'i for the first time in 2017

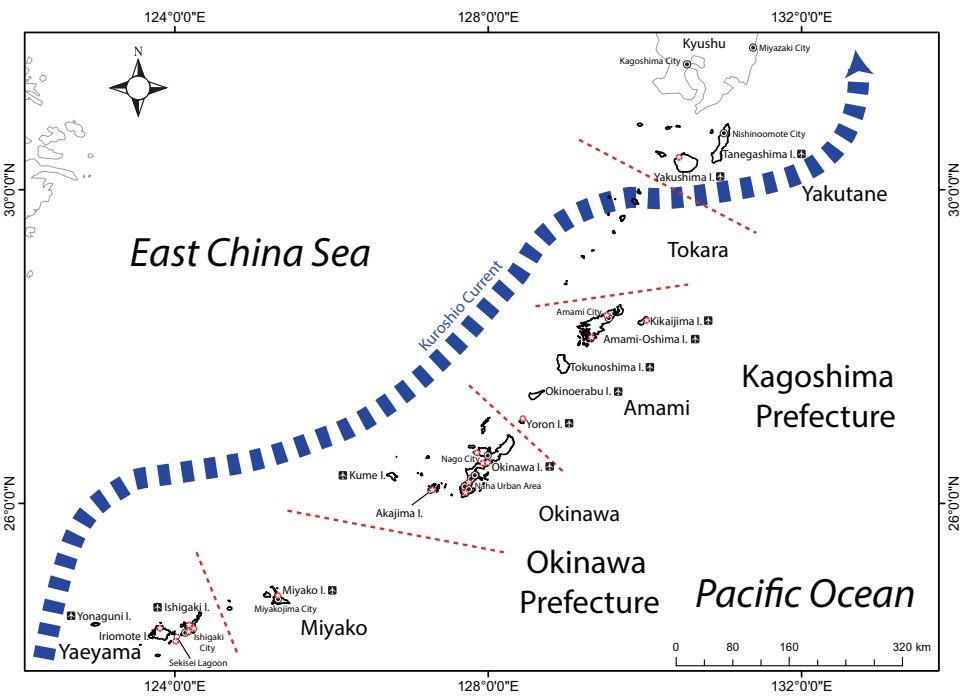

**Figure 1  Map of the Ryukyu Islands (RYS) with sub-regions examined in this study, and relevant geographic features and research institutions (red stars).** Airplane icons next to island names indicate presence of airport.

(total number 9,579,900; *Ryukyu Shimpo, 2018*; also FY2017 data on Okinawa Prefecture homepage http://www.pref.okinawa.jp/site/bunka-sports/kankoseisaku/h28nendo.html).

Although the RYS contain high levels of marine species diversity, until now there has been no marine biodiversity overview that covers the archipelago in detail (but see *Fujikura et al. (2010)*'s general overview of marine biodiversity of Japan with a focus on Sagami Bay). Here, we conduct an extensive English language literature data-mining review to provide information and conduct a gap analysis on the status of marine biodiversity research within the RYS, with specific information on six sub-regions within the RYS for six important and representative marine taxa (Pisces, Crustacea, Echinodermata, Cnidaria, Porifera, Dinoflagellata). Furthermore, we review and compare data of ecological studies in the RYS to those of other major reef regions (Caribbean, Great Barrier Reef, Red Sea). Finally, we discuss and highlight the trends of biodiversity related research in the RYS, emphasizing the need for continued research as the data gaps hamper our understanding of marine biodiversity and conservation efforts in this important coral reef region.

# MATERIALS AND METHODS

## The Ryukyu Islands (RYS)

We divided the RYS into six sub-regions based on geographical, historical, and administrative information (Table 1); the sub-regions generally follow those in Coral Reefs of Japan (*Nishihira, 2004*) and as used by various levels of Japanese government.

Reimer et al. (2019), *PeerJ*, DOI 10.7717/peerj.6532

**Table 1  Information on the six sub-regions investigated in this study in the Ryukyu Islands (RYS).**

| Sub-region | Yakutane | Tokara | Amami | Okinawa | Miyako | Yaeyama | RYS total | Reference(s) |
|---|---|---|---|---|---|---|---|---|
| **Major islands** | Yakushima, Tanegashima, Kuchinoerabu, others | Nakanoshima, Suwanose, Kuchi-noshima, Taira, Takara, Kodakara, Akuseki, others | Amami-Oshima, Kikai, Tokunoshima, Okinoerabu, Yoron, others | Okinawa Main Island, Kume, Izena, Iheya, Kerama Islands, Ikei Islands, Aguni, Ie, Sesoko, Kouri, others | Miyako, Ikema, Tarama, others | Ishigaki, Iriomote, Taketomi, others | 198 islands (not including <0.01 km$^2$) | NA |
| **Notable marine research stations & institutes** | Yakushima Umigane-kan | None | Kagoshima University Amami Station, Seikai National Fisheries Research Institute Amami Station, Kikai Coral Reef Research Institute (from 2014), Kagoshima U. Fac. Fisheries Yoron Station | Akajima Marine Science Laboratory (closed 2017), Okinawa Institute of Science and Technology (from 2011), Okinawa National College of Technology, University of the Ryukyus Tropical Biosphere Research Center Sesoko Station, University of the Ryukyus Main Campus, Itoman Pref. Exp. Center, Meio University | Miyakojima City Museum | University of the Ryukyus Iriomote Field Station, Ishigaki Pref. Exp. Station, Ishigaki MoE Parks Station, Kuroshima Sea Turtle Station | | Various institution homepages |
| **OBIS cover (# squares)** | 27 | 14 | 36 | 50 | 11 | 26 | 164 | OBIS |

Reimer et al. (2019), *PeerJ*, DOI 10.7717/peerj.6532

**Table 1** (*continued*)

| Sub-region | Yakutane | Tokara | Amami | Okinawa | Miyako | Yaeyama | RYS total | Reference(s) |
|---|---|---|---|---|---|---|---|---|
| **Land area (km²)** | 1,030 | 101.35 | 1,231.47 | 1,418.59 | 226.5 | 587.16 | 4,595.07 | *Geospatial Information Authority of Japan (2017)* |
| **Population** | 41,870 | 768 | 107,238 | 1,341,553 | 52,456 | 54,092 | 1,597,977 | *Kagoshima Prefecture (2017)*, *Okinawa Prefecture (2018)* |
| **Population density (/km²)** | 40.7 | 7.6 | 87.1 | 945.7 | 231.6 | 92.1 | 347.8 | NA |
| **Geological formation** | Volcanic, sedimentary, granite uplift | Volcanic, coral reefs | Volcanic, sedimentary, uplift, coral reefs | Volcanic, sedimentary, uplift, coral reefs | Volcanic, coral reefs | Volcanic, sedimentary, uplift, coral reefs | N/A | *Kayanne, Hongo & Yamano (2004)*, *Fujita et al. (2015)* |
| **Annual average SST (°C)** | | 24.3 | 24.5 | 25.0 | 25.8 | 25.2 | N/A | *Hasegawa & Yamano (2004)*, *Kajiwara & Matsumoto (2004)*, *Nakai & Nojima (2004)*, *Nakai & Oki (2004)*,*Sakai (2004)*, *Shimoike (2004)*, *Yokochi (2004a)* |
| **Reef perimeter (km)** | Local coral reef flats only | 19 | 420.3 | 382.2 | 121.6 | 268.4 | 1,211.5 | *Hasegawa & Yamano (2004)*, *Kajiwara & Matsumoto (2004)*, *Nakai & Nojima (2004)*, *Nakai & Oki (2004)*,*Sakai (2004)*, *Shimoike (2004)*, *Yokochi (2004a)* |

Reimer et al. (2019), *PeerJ*, DOI 10.7717/peerj.6532

**Table 1** (*continued*)

| Sub-region | Yakutane | Tokara | Amami | Okinawa | Miyako | Yaeyama | RYS total | Reference(s) |
|---|---|---|---|---|---|---|---|---|
| **Coral community area (ha)** | 118 | | 5,951.2 | 6,980 | 1,957.1 | 19,231.5 | 34,237.8 | *Hasegawa & Yamano (2004)*, *Kajiwara & Matsumoto (2004)*, *Nakai & Nojima (2004)*, *Nakai & Oki (2004)*, *Sakai (2004)*, *Shimoike (2004)*, *Yokochi (2004a)* |
| **Fishing activities** | Recreational, commercial | Recreational, some commercial | Recreational, commercial | Recreational, commercial | Recreational, commercial | Recreational, commercial | N/A | *Hasegawa & Yamano (2004)*, *Kajiwara & Matsumoto (2004)*, *Nakai & Nojima (2004)*, *Nakai & Oki (2004)*, *Sakai (2004)*, *Shimoike (2004)*, *Yokochi (2004a)* |
| **Agricultural activities** | Sugarcane, rice, sweet potatoes, vegetables, other fruits, flowers, tobacco, others | Minimal | Sugarcane, pineapple, potatoes, vegetables, other fruits, flowers, tobacco, others | Sugarcane, vegetables, pineapple, other fruits, flowers, tobacco, others | Sugarcane, vegetables, other fruits, flowers, tobacco, others | Sugarcane, vegetables, pineapple, other fruits, flowers, tobacco, others | N/A | Kagoshima, Okinawa prefectural homepages |
| **Other activities/issues** | Local tourism | Minimal | Local tourism | Extensive tourism, red soil runoff, landfill, military bases local pollution & eutrophication | Extensive tourism | Extensive tourism, red soil runoff, landfill | N/A | *Nakano (2004a)*, Kagoshima, Okinawa prefectural homepages |

Reimer et al. (2019), *PeerJ*, DOI 10.7717/peerj.6532

**Table 1** (*continued*)

| Sub-region | Yakutane | Tokara | Amami | Okinawa | Miyako | Yaeyama | RYS total | Reference(s) |
|---|---|---|---|---|---|---|---|---|
| **COTS out-breaks** | None | None | 1970s onwards | 1970s onwards | 1957–59, 1970s–1980s, 2004~ | 1970s–1980s, 2007~ | N/A | *Yokochi (2004b)*, Ministry of Environment |
| **Water quality notes** | Oligotrophic oceanic | Oligotrophic oceanic | Oligotrophic oceanic with turbid bays | Oligotrophic oceanic with turbid bays, local pollution & eutrophica-tion | Oligotrophic oceanic with turbid bays | Oligotrophic oceanic with turbid bays | N/A | *Hasegawa & Yamano (2004)*, *Kajiwara & Matsumoto (2004)*, *Nakai & Nojima (2004)*, *Nakai & Oki (2004)*, *Sakai (2004)*, *Shimoike (2004)*, *Yokochi (2004a)* |
| **Recent years of level 2 bleaching events** | 1998, 2001, 2016–7 | 1998 | 1998, 2016–7 | 1998, 2001, 2016–7 | 1998, 2001, 2016–7 | 1998, 2001, 2010, 2016–7 | N/A | *Nakano (2004b)*, NOAA, Ministry of Environment |
| **National parks** | Yakushima | None | Amamigunto | Yanbaru, Keramashoto | None | Iriomote-Ishigaki | 5 parks | *Hasegawa & Yamano (2004)*, *Kajiwara & Matsumoto (2004)*, *Nakai & Nojima (2004)*, *Nakai & Oki (2004)*, *Sakai (2004)*, *Shimoike (2004)*, *Yokochi (2004a)*, Ministry of Environment |
| **Number of coral species** | 151 | >151 | 200 | 340 | 302 | 363 | N/A | *Nishihira & Veron (1995)*, *Nishihira (2004)* |

The six sub-regions (south to north) are the island groups of Yaeyama, Miyako, Okinawa, Amami Oshima, Tokara, and Yakutane. The first three sub-regions are within Okinawa Prefecture, while the last three are within Kagoshima Prefecture, and are as follows:

a. Yaeyama Islands: the southernmost group of islands in the RYS, this group experiences the most tropical conditions, has the most well-developed coral reefs (*Nishihira, 2004*), including the Sekisei Lagoon, Japan's largest reef system, and is generally thought to have the highest biodiversity within the entire archipelago (*Nishihira & Veron, 1995*; *Roberts, Mittermeier & Schueler, 2002*; Table 1). This sub-region includes the major islands of Ishigaki and Iriomote as well as several smaller islands.

b. Miyako Islands: includes the large island of Miyako as well as several surrounding smaller islands. This sub-region is notable for having a coral reef system with extensive cave systems and endemic species (e.g., *Shimomura, Fujita & Naruse, 2012*).

c. Okinawa Main Island and region: this sub-region is dominated by Okinawa Main Island, the largest and by far the most populous island in the RYS (Table 1). In addition, the island is surrounded by numerous smaller islands notable for their relatively pristine condition and protection within two national parks.

d. Amami Oshima Island and region: Amami Oshima is the second largest island in size and population in the RYS, but this region also includes other major islands such as Yoron, Okinoerabu, and Tokunoshima, as well as many smaller island groups. Notable for endemic terrestrial species, the marine life of this subregion is thought to be understudied when compared with regions further south (e.g., *Fujii, 2016*; *Nakae et al., 2018*). The southernmost portion of Kagoshima Prefecture, this area was historically sometimes included within the former Ryukyu Kingdom (current Okinawa Prefecture).

e. Tokara Islands: the smallest and least populated sub-region within the RYS, this group is often considered part of the Yakutane Islands, but differs in several important ways, as it has more developed coral reefs than areas further north in the Yakutane sub-region and south around Amami Oshima (*Nakai & Nojima, 2004*), and is heavily influenced by the Kuroshio Current. This sub-region consists of 12 small islands stretched across 160 km, with six islands having well-developed coral reefs (*Nakai & Nojima, 2004*). As the least developed sub-region, this area, unlike all other sub-regions, is not easily reachable by major air transport systems, and is considered the least well-studied area in the RYS.

f. Yakutane Islands (also known as the Osumi Islands): consisting of the two major islands of Yakushima and Tanegashima along with neighboring smaller islands, the Kuroshio takes a sharp turn to the east south of Yakushima. This sub-region is considered the northern limit of modern coral reef development in the region (*Nakai & Nojima, 2004*) and the northern limit of the subtropical region of Japan.

### Web of science taxa and sub-regions search

We searched within the Web of Science (WoS) for English language papers on six representative marine taxa within the RYS; Pisces, Mollusca, Crustacea, Echinodermata, Cnidaria, and Dinoflagellata, utilizing search strings (Table S1). We determined the sub-region location of each paper of these six taxa within the WoS based on the title, key words, and abstract information. When the title and abstract only contained "Okinawa", "Ryukyu", or "Nansei", with no further information, we categorized these

papers as "Ryukyu/Nansei unspecified", as "Okinawa" may refer to the entire Okinawa Prefecture, and "Ryukyu" and "Nansei" may refer to anywhere within the RYS island chain. Additionally, deep-sea publications were not included in our examinations. Publication numbers were compiled for 1995–2017 for each taxon for each sub-region to examine what taxa have been investigated in what sub-region. All information utilized was obtained from within the WoS search tool. The search was conducted in August/September 2017.

## Web of science ecology search and comparison

We searched eight principal topics in ecological studies ("apex predators", "connectivity", "coral bleaching", "coral reproduction", "herbivory", "marine protected areas", "Porifera", "reef-associated bacteria") within WoS following the search strings utilized by *Berumen et al. (2013)* in their review on biodiversity research in the Red Sea (see also Table S1). Subsequently, we compared the data for four reef regions across the globe. The regions and search strings used to filter the data were the following: RYS (search string was "Ryukyus*" OR "Nansei" OR "Okinawa*"), Caribbean (CAR; search string was "Caribbean"), Great Barrier Reef (GBR; search string was "Great Barrier Reef"), and the Red Sea (RES; search string was "Red Sea"). Publication numbers were compiled annually (1995–2016) and by ecology topic (as above). The search was conducted on September 20, 2017.

## Ocean biogeographic information system search

We also searched the six sub-regions of the RYS within the Ocean Biogeographic Information System (OBIS, *OBIS, 2017*) for the six representative marine taxa (Cnidaria, Crustacea, Dinoflagellata, Echinodermata, Mollusca, and Pisces) with the aim of examining spatial differences of readily available online data for these taxa within sub-regions. Using the highest grid resolution of OBIS (0.1 degree), we examined all square grids that covered the coastline of each island of the RYS and noted the number of records for each quadrat for each taxon. The number of quadrats examined in each sub-region ranged from 11 in Miyako to 50 in Okinawa (Table 1). The search was conducted in August 2017.

## RESULTS

### Web of science taxa and sub-regions search

In total, from our WoS searches for English language papers between 1995 and 2017, we examined 980 papers, which contained information for 1,023 sub-region occurrences (some papers had less than one sub-region in their content). Of these occurrences, 420 were from the Okinawa Main Island sub-region, 307 from an unspecified area in the RYS, 199 from Yaeyama, 48 from Amami Oshima, 29 from Yakutane, 16 from Miyako, and 4 from Tokara (Fig. 2).

By taxa, the groups Pisces ($n = 346$), Cnidaria ($n = 233$), and Crustacea ($n = 225$) had the most occurrences, with all other groupings <100 occurrences (Mollusca $n = 92$, Echinodermata $n = 51$, Dinoflagellata $n = 44$; Fig. 2). Of note was the fact that ~80% of both Echinodermata and Dinoflagelleta papers were from Okinawa (40 of 51 papers, 36 of 44, respectively). Papers dealing with Pisces were most numerous for Yakutane ($n = 12$),

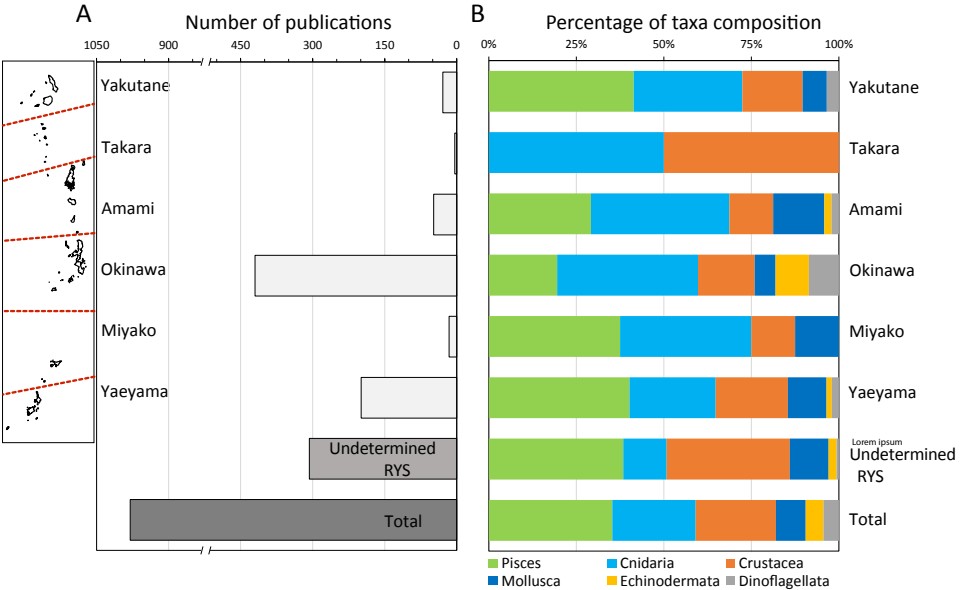

**Figure 2 Total number of publications in the Web of Science, and breakdown of these publications by marine taxa.** (A) Total number of publications in the Web of Science (1995–2017) for six different marine taxa (Pisces, Crustacea, Echinodermata, Cnidaria, Porifera, Dinoflagellata) for each sub-region in the RYS as well as for undetermined locations and overall total, and (B) the breakdown of these publications by marine taxa. Reference map of the RYS included on the left side of (A).

Okinawa ($n = 82$), and Yaeyama ($n = 80$), while papers on Cnidaria were most numerous for Amami ($n = 19$) and Okinawa ($n = 169$), and Cnidaria and Crustacea were equally numerous for Tokara ($n = 2$ each) and Miyako ($n = 6$ each). For unspecified sub-regions, Pisces ($n = 118$) and Crustacea ($n = 108$) were the most numerous taxa (Fig. 2).

## Web of science ecology search and comparison

Our WoS search results showed that the RYS had fewer publications overall ($n = 1,288$; Fig. 3) when compared to the three other coral reef regions examined for the same time period (GBR $n = 6,242$, CAR $n = 6,990$, RES $n = 4,493$). Additionally, RYS publication numbers were lower for all eight ecological topics analyzed (Fig. 4). In particular, numbers for RYS were comparatively very low for herbivory and marine protected areas (Figs. 4B and 4D, respectively). Temporally, the number of papers published for all regions increased noticeably between 1995 and 2016 (Fig. 3), with the number of RYS papers increasing approximately 4.0 times (1995 $n = 24$ publications vs. 2016 $n = 97$), CAR papers increasing 4.6 times (1995 $n = 100$ vs. 2016 $n = 460$), GBR papers increasing approximately 7.6 times (1995 $n = 66$ vs. 2016 $n = 504$), and RES papers increasing approximately 8.4 times (1995 $n = 47$ vs. 2016 $n = 397$).

## Ocean biogeographic information system results

OBIS results examining the numbers of records of different taxa within the sub-regions showed great variation, with some general trends appearing. In general, the three more northern sub-regions within Kagoshima Prefecture had fewer records than those in

a

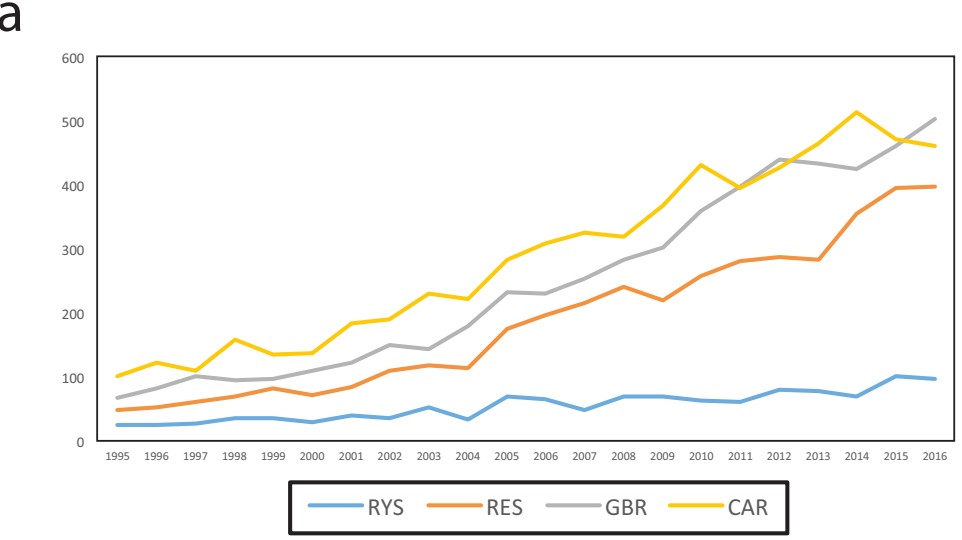

b

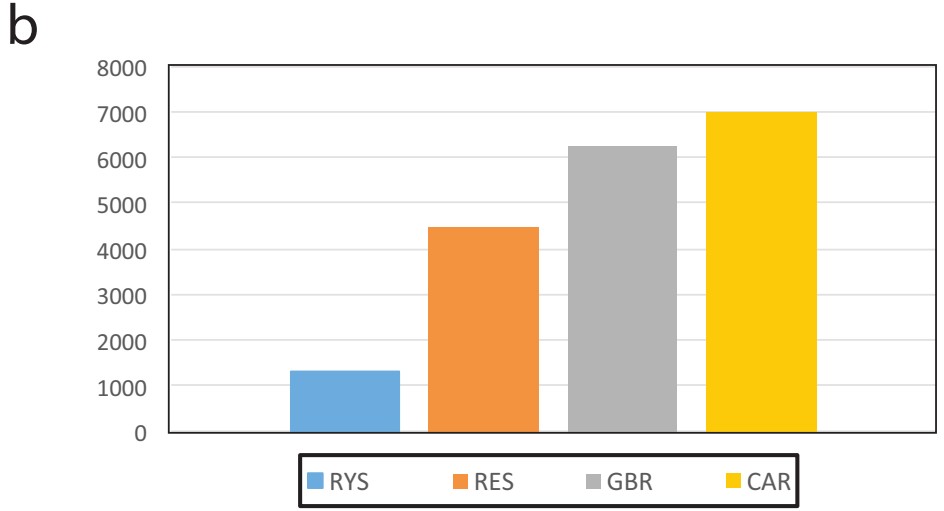

**Figure 3** (A) Numbers of ecological publications per year, and (B) the total number of publications for the Ryukyus (RYS; blue), Red Sea (RES, orange), Great Barrier Reef (GBR, grey), and Caribbean (CAR, yellow) from 1995 to 2016 in the Web of Science.

Okinawa Prefecture for Cnidaria, Crustacea, Echinodermata, Mollusca, and Pisces. Within Okinawa Prefecture (and the RYS), Okinawa consistently had the highest numbers of records, with the highest numbers observed around Akajima (Crustacea, $n = 200$–$500$) and the west coast of Okinawa Main Island (Cnidaria, Crustacea, Echinodermata, Mollusca, and Pisces). Conversely, even within the Okinawa sub-region, some areas such as the northeast coast of Okinawa Main Island had none or only few records (Fig. S1a). Additionally, there was only one record for the entire RYS within OBIS for Dinoflagellata in shallow water (Fig. S1b).

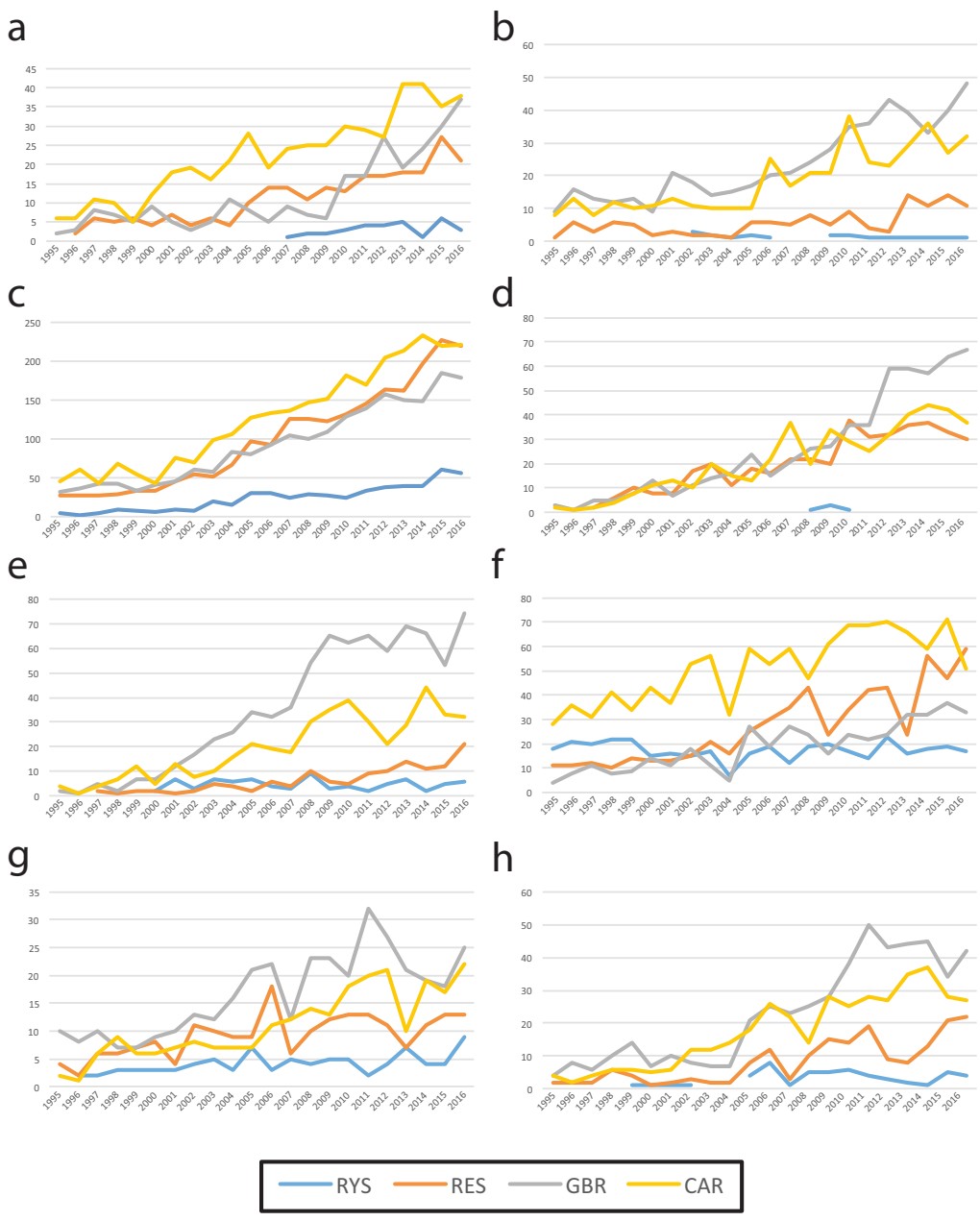

**Figure 4  Number of ecological publications per year for four regions from 1995 to 2016 in the Web of Science; the Ryukyus (RYS; blue), Red Sea (RES, orange), Great Barrier Reef (GBR, grey), and Caribbean (CAR, yellow) by topic.** (A) Apex predators, (B) herbivory, (C) connectivity, (D) marine protected areas, (E) coral bleaching, (F) Porifera, (G) coral reproduction, and (H) reef-associated bacteria.

## DISCUSSION

From the WoS and OBIS analyses of sub-regions and taxa occurrences it became clear, as in many other marine regions (*Hughes, Bellwood & Connolly, 2002*), that serious taxonomic and geographic biases are present in marine research in the RYS. Some of this taxonomic

bias may stem from the commercial importance of Pisces and Crustacea in Japan, which has resulted in many studies on various species' lifecycles and aquaculture methodologies (e.g., *Hamasaki & Kitada, 2008*; *Nanami et al., 2010*). Research on these topics, while often conducted somewhere in the RYS, generally did not include field observations or sampling information as the focus was more on *ex situ* analyses and model species. This was reflected in these two groups' dominance of the "unspecified RYS sub-region" category as no specific location within the RYS was given within these papers (Fig. 2). Another source of taxonomic and geographic bias likely stems from the presence of specialist researchers, and this was more noticeable in relatively understudied Echinodermata and Dinoflagellata, with one researcher each based in the Okinawa sub-region contributing >20% of the WoS research on each taxa (Echinodermata, T Uehara, 20/50 papers = 40%; Dinoflagellata, S Suda, 9/44 papers = 20.5%; both University of the Ryukyus).

Overall, most work in the RYS has been conducted on Pisces, Crustacea, and Cnidaria, with the large majority (57.94%, $n = 135/233$) on Scleractinia hard corals, and surprisingly far less work on other commercially important groups such as Echinodermata and Mollusca. While Mollusca research was somewhat evenly spread around the RYS, approximately 80% of Echinodermata research was conducted in the Okinawa sub-region, again due to the presence of a specialist based in the region (T Uehara, 20/50 papers = 40%; also Fig. 2B). Due to recent commercial pressure and reported large drops in abundances of some echinoderms (e.g., annual Okinawa Prefecture harvest of *Tripneustes gratilla* sea urchin from >700 t in 1976 to <50 t in late 1990s (*Kakuma, 2003*), see also *Soliman, Fernandez-Silva & Reimer (2016)*; *Soliman et al. (2016)*), it is clear that more research is urgently needed in other sub-regions. *Hughes, Bellwood & Connolly (2002)* suggested nearly two decades ago that more work is needed on understudied small and cryptic coral reef taxa in understudied locations, and this is still true for the RYS.

From the analyses of records of various taxa in OBIS, we can gain an indication of research patterns within each sub-region. While generally understudied sub-regions such as Tokara had a lack of occurrence records for all taxa across all areas inside the sub-region, in the case of more well-studied sub-regions, these areas were often directly adjacent to marine research stations (e.g., 200–500 Crustacea occurrence records on Akajima, containing Akajima Marine Station, active until 2017; Fig. S1a) and have many more occurrence records than in other areas. Thus, while Okinawa and Yaeyama can be considered to be comparatively well-studied inside RYS, there are areas within both sub-regions that are almost completely uninvestigated. As conservation studies require occurrence data across species ranges, research on these uninvestigated areas are a necessity. Additionally, the presence of marine research stations is a driving force for research, and this is demonstrated by the OBIS records for the Miyako sub-region, which despite a relatively large human population, has no research-focused marine station (Table 1), and a corresponding general lack of scientific publications and data available (e.g., Fig. 2).

The WoS does not include all scientific publications from each region of the world as it does not index all scientific journals, and its coverage in some fields is less complete than in others. The lack of WoS coverage is particularly acute when examining English language marine science publications from Japanese waters. Japan has a long history of marine
biodiversity and coral reef science (e.g., *Kawaguti, 1940*), and even today much research is published in Japanese, the large majority of which is in journals that do not appear in the WoS or any other central online database. An exception is Nippon Suisan Gakkaishi, and even though it appears in the WoS, some articles in this journal list title and authors only, with no abstract available in English, and the journal even occasionally contains articles with no English at all. Such domestic journals are held in high regard in many scientific fields within Japan, including marine and fisheries sciences, and contain much valuable and important data. Our initial analyses of Japanese language literature suggest a strong bias towards fisheries (Pisces, some Crustacea) in the RYS (data not shown). Failure to access these journals and their contents undoubtedly results in not gaining a complete picture of marine sciences in Japan, including our examination here of marine biodiversity in the RYS. We suggest that Japanese language science publications make the effort to include translations of the title, authors, and abstract to allow more access from the international science community, as is already performed by such journals as Nippon Suisan Gakkaishi (for most articles, in the WoS) and Fauna Ryukyuana (for all articles, not in the WoS). Additionally, for aquaculture or model species studies, listing the exact location from where specimens were collected would be helpful for mapping records and distributions of species in the Oceanographic Biology Information System (*OBIS, 2017*) or other databases. Inclusion of complete datasets online in repositories or as supplementary material would also greatly aid in occurrence record mapping.

From the WoS search on ecological topics, the relative and comparative lack of research in the RYS compared to the 'major' coral reef areas of CAR, GBR, and the RES is apparent. Our results were somewhat expected, particularly given the relatively small size of the RYS (approximately 4,642 km$^2$ area and c. 1,200 km in length) in comparison to the GBR, CAR and RES (17,400 km$^2$ area c. 2,300 km length; 10,530 km$^2$ area; 8,890 km$^2$ area c. 2,000 km length; respectively, data from *Berumen et al. (2013)*); the coral reefs of the RYS are approximately half the size of the next-smallest region RES. However, the amount of research conducted in the area is not a direct function of the size of coral reefs in any region. On the other hand, in terms of human populations immediately adjacent to reefs, the RYS (Table 1) could be considered to have higher numbers than those of the GBR or even the RES, particularly given that the other three regions have continental landmasses much larger than those of the RYS. The RYS also have an abundance of marine research institutions (Fig. 1, Table 1).

When examining the trends for the different ecological topics, the deficiencies of research in the RYS become starkly clear, with almost no research conducted on ecosystem sciences such as herbivory, or on marine protected areas (Fig. 4). Historically, Japan and Okinawa have been somewhat slow to adopt marine conservation measures with legal strength (*Reimer et al., 2015*), and there are still no no-take zones in the RYS. It also appears from our results that scientists based in the region have been equally slow to adopt research on these topics, despite a public need for such third-party research given the controversy over continuing coastal development in Okinawa (*McCormack, 1999*; *Hook, 2010*). Additionally, and somewhat surprisingly, there has been little research on apex predators, despite high public interest in Japan in this group (e.g., world-famous shark

displays at Churaumi Aquarium in Okinawa, visited by >3,000,000 people/year; *Churaumi Aquarium, 2018*). Given the high rates of marine endemism and biodiversity in this region (*Roberts, Mittermeier & Schueler, 2002*), more efforts should be made to conduct research on these topics in the RYS.

It is well known that Japan has a strong fisheries industry, with much funding for such research (e.g., Japan Fisheries Research and Education Agency created in 2016), and many national universities include a Faculty of Fisheries (e.g., Kagoshima University, national university for Kagoshima Prefecture containing Yakutane, Tokara, and Amami island regions). Thus, there may be a general bias in research and instruction towards fisheries-related species, possibly at the expense of other marine biodiversity research and taxa. More capacity building and investment are needed in other fields, such as non-fisheries-related marine biodiversity, marine ecology, and marine conservation, as our results demonstrate.

The amount of scientific papers being published is known to be increasing over time (*Bornmann & Mutz, 2015*), and marine science is no exception. However, worryingly, the speed at which marine scientific research in the RYS has increased has not kept pace with the other three regions we examined (Fig. 3). While numbers of publications from the RYS (and all other regions) are increasing, given the large number of coral reef, fisheries, and marine science researchers in Japan (e.g., the Japanese Coral Reef Society created in 1997 has over 600 members (JCRS homepage http://www.jcrs.jp)), we expected the gap between the RYS and other regions to be smaller. At current rates and based on these data from the past twenty-one years, compared to other regions the RYS are less studied now than in 1995.

## CONCLUSIONS

Marine biodiversity and ecology research in the RYS, while steadily advancing, lags behind the progress of other major coral reef regions in the world. In particular, research levels on conservation topics are dramatically lower than in other coral reef regions, despite the need for conservation and protection of these ecosystems (Fig. 4). Additionally, despite the large amount of marine research infrastructure including numerous research facilities and a large population base, and despite the comparatively small area of the RYS (Table 1), there are taxa in both sub-regions and smaller areas within sub-regions that are almost completely unstudied. Moving forward, local, prefectural, and national governments and stakeholders should focus on addressing the gaps in our knowledge base. Marine biodiversity research in the RYS will benefit from continuing to build international collaborative research projects to broaden the breadth of taxonomic studies and also the scope of ecosystem-based and conservation planning studies. Such works combined with a more robust legal framework and the establishment of functioning no-take and marine protected areas should be able to better conserve and protect RYS coral reef ecosystems and their valuable ecosystem services for future generations (Table 2).

**Table 2  Summary of key research priorities to support marine biodiversity, marine ecology, and marine conservation in the Ryukyus (RYS).**

| Research priority | Cause(s)/source(s) of need | Achievability rank (1 (low) — 5 (high)) + comments |
|---|---|---|
| More research on Yakutane, Miyako and Tokara island groups. | Low numbers of publications. | 4. Dependent on funding to organize expeditions. |
| More research on Dinoflagellata. | Least studied taxon ($n = 44$), of ecological interest + importance. | 3. Requires taxonomists and chemical and molecular equipment. Some taxa impossible to keep in culture. |
| More research on Echinodermata. | Understudied ($n = 51$). Most studies only around Okinawa Island. Includes coral predators and species of economic interest, potentially threatened. | 4. Requires taxonomists and molecular equipment. |
| More research on Mollusca. | Relatively understudied ($n = 92$). Includes coral predators and species of economic interest, potentially threatened. | 4. Requires taxonomists and molecular equipment. |
| More studies on 'herbivory' and 'apex predators'. | Important in marine conservation. Understudied in the RYS. Herbivores and apex predators potentially threatened/overharvested. | 5. Requires scientists focused on these topics. |
| More studies on marine protected areas (MPAs). | Critical for marine conservation. | 2. Requires functioning MPAs. In Japan, no-take-zones are currently absent. |
| Implementation of new MPAs. | Critical for marine conservation. | 1. MPAs are still largely seen negatively. More effort should be invested in public outreach and education. |
| Making Japanese literature more accessible. | Science should be accessible by international scientific community. Possibility of increased international collaboration and funding. | 5. Requiring at least title and abstract submitted in English should be feasible for scientific journals. Some journals already do this. Encourage publishing in English. Research institutions should provide language support. |

## ACKNOWLEDGEMENTS

Data in this study were generated as part of a doctoral-level class entitled "Advanced Marine Biodiversity", taught by James Davis Reimer in 2017, and part of the Okinawa International Marine Science Program (OIMAP) at the Graduate School of Engineering and Science at the University of the Ryukyus (UR). This work was partially inspired by a Red Sea biodiversity research overview by *Berumen et al. (2013)*. This article was prepared as a class project by the members of a PhD program with all authors contributing equally to produce the research and the end product of the article. They have also been named in alphabetical order to reflect their equal contributions. We thank Drs. T Naruse (UR) and T Fujii (Kagoshima University) for information on marine research institutes in the RYS. Three reviewers greatly improved an earlier version of this manuscript.

### Funding

Xuan Hoa Nguyen was supported by a Vietnamese Government (911) scholarship, Maria E.A. Santos, Hin Boo Wee, and Yee Wah Lau were supported by Japanese Government (MEXT) scholarships, and Piera Biondi was supported by a Mitsubishi scholarship. The

funders had no role in study design, data collection and analysis, decision to publish, or preparation of the manuscript.

## Grant Disclosures

The following grant information was disclosed by the authors:
Vietnamese Government (911) scholarship.
Japanese Government (MEXT) scholarships.

## Competing Interests

James D. Reimer is an Academic and Section Editor for PeerJ.

## Author Contributions

- James Davis Reimer, Piera Biondi, Yee Wah Lau, Giovanni Diego Masucci, Xuan Hoa Nguyen, Maria E.A. Santos and Hin Boo Wee conceived and designed the experiments, performed the experiments, analyzed the data, prepared figures and/or tables, authored or reviewed drafts of the paper, approved the final draft.

## Data Availability

Our raw data are available in Table S1.

## Supplemental Information

Supplemental information for this article can be found online at http://dx.doi.org/10.7717/peerj.6532#supplemental-information.

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
