# Peer review of "Marine biodiversity research in the Ryukyu Islands, Japan: current status and trends"

_PeerJ, doi:10.7717/peerj.6532_

## Round 0.1 · original submission · Major Revisions

As you will see, the three reviewers took somewhat different approaches in their discussions of your manuscript. I especially agree with Reviewer 1 that it would benefit from a clearer treatment of the Japanese-language publications and with Reviewer 3 who would like to see a more concise statement of the significance of your findings and the placing of your results in the context of prior research. There are a lot of data provided in your interesting paper -- it would be very helpful if you could find a way to clearly highlight the evidence for the taxonomic and regional bias that was found.

·

Basic reporting

The article describes the relative paucity of research in the RYS, highlighting under-studied sub-regions, taxa and ecological topics. I enjoyed reading it, thank you!

The article is generally clear and well written. There are a few grammatical errors/typos and the authors should check for these more carefully. Ones I spotted are:
Line 20: change to ‘spanning protists to vertebrates’
Line 58: across
Line 176: change ‘less’ to ‘fewer’

Figure 1. I wonder if this figure might be clearer if the authors split it into two, i.e. Figure 1a would be the map and Figure 1b the pie charts. It’s not a huge issue but it feels a bit cluttered to me. Pie charts are also notoriously difficult to assess by eye apparently although I appreciate your choice since it allows the reader to simultaneously visulalise the number of publications. Another option could be a barplot similar to ggplot’s “geom_bar”.

Unfortunately I was not able to open the Supplementary eps file. I apologise - I should have checked earlier and have now run out of time to get the file in a different format.

Experimental design

The authors provide an overview of the state of marine biodiversity research in the Ryukyu Archipelago (RYS), investigating sub-region, taxa and ecological topic. Additionally they look at the spatial bias of records with respect to marine research stations and institutes.
The research is well defined and methods well described.

The authors give considerable discussion to the problem of a sizeable number of research articles published solely in Japanese. These articles are not considered in their study. It is therefore hard to know whether the RYS is genuinely as under-studied as their results suggest. I see two options here. The first, which would be preferable, would be for the authors to extend their study to the Japanese language publications. I appreciate that if these journals are not databased this could be somewhat laborious – and would also of course require fluency in Japanese and access to the journals. However, I have manually searched journals not included on WoS myself for articles relating to a particular region (in this case Sundaland) and it actually only took a few days. Adding in these Japanese language articles would add real value to the study. It would be interesting – and important - to know whether these articles contain the same biases as English-language articles and also what proportion of research on the RYS is unavailable to a global audience.

The second, easier option, would be to be much more upfront that the articles are English-language. So, beginning in the abstract and continuing throughout, ‘this study provides an overview of the state of English-language marine biodiversity research in the RYS’ or similar.

It may be that a compromise between the two options is possible, e.g. count the number of Japanese publications that relate to the RYU without going into detail on ecological topic. However, I would still prefer the first option.

Validity of the findings

See my comments on the English/Japanese language publications above. That aside, the data appear robust and conclusions well stated.

·

Basic reporting

The paper is very well written - there are however a small number of places where sentence structure could be adjusted to achieve greater clarity.
Examples from the abstract:
First line ... Could be reworded as "Coral reef of the subtropical Ryukyu Archipelago (RYS; also known as the Nansei Islands) in Japan have been shown to harbour very high levels of marine biodiversity".

Last line... Could be reworded into two senteces i.e. "Coral reefs in the RYS feature high levels of both endemism and anthropogenic threats, subsequently they contain a concentration of some of the worlds most critically endangered marine species. To protect these demonstratably threatened species, more data is needed to fill the research gaps identified in this study.

When you reference to 4.0 times (rate of increase) could this be expressed as 4-fold?

Please consider adding an additional sentence at the front end of the abstract to justify the significance of biodiversity in the RYS and to set up why its important to explore the extent of reserach and to identify gaps. In order to fit that sentence it I would encourage revisting the rest of the abstract with the view of writing it a bit more succintly.

Intro
line 58 - spelling error "across"
line 62 - please split into two sentences.
line 64 - could be rewritten for clarity. e.g. "In 2002, a global analysis of regional conservation priority ranked southern Japan and Taiwan first due to the high levels of both endemism and threats, highlighting that rapid conservation action is needed".
line 73 - can you state the numbers of visitors in each region for context?
line 76 - any refs to substantiate the diversity in RYS

Experimental design

The paper appears to be a "gap analysis" - have the authors considered using this phrase as it may help further conceptualize the entire study.

For the benefit of audiences outside of Japan, is there any way you can include a satellite image of the RYS - potentially showing well known places such as Tokoyo and Okinawa and places of interest that are mentioned in text such as Sekisei Lagoon or some of the other marine field stations mentioned in Table 1.

I note data us used from WoS and OBIS - I am aware in my region the OBIS records are very scant and there is much data in museum databases that has not been included in OBIS to date - how utilized is the OBIS dbase for this region and have you considered exploring releveant museum databases or additional species records?

Validity of the findings

All the findings are sound but I would encourage the authors to include some additional discussion points. For example - does the higher number of fish, cnidaria and crustacean research papers reflect there is a bias towards these areas that stems from these being the areas where there is local and national academic expertise? Hence this has led to teaching and learning biases within Japanese universities? If so, does that mean more capacity building and investment is needed in other subject areas?

How many of the papers include international collaborations? I dont expect the authors to go back and recalculate this across the 960 papers but it could be worth mentioning that the current global landscape of research direction is constantly moving towards collaborative, multi-institutional projects - hence in order to address the knowledge gaps that have been identified here, it could be worthwhile making a statement such as ... Marine biodiversity reserch in the RYS will benefit from continuing to build international collaborative research projects to broaden the breadth of taxonomic studies and scope of ecosystem based and conservation planning studies....

I am curious to know how well utilized OBIS is - there is definatlely the opportunity here to further encourage the scientific community to utilize this dbase.

line 257 - please quantify the term "higher numbers" when referring to population density between GBR, RES and RYS.

It might be worth mentioning some of the other "non-charasmatic or cryptic" taxa that are not specifically examined in this paper to highlight the extent of marine biodiversity and how existing research is just stratching the surface. Also the Cnidaria is a huge phylum is it worth mentioning any biases within - such as bias towards Anthozoa or hard corals over other classes/subclasses/families?

Is it worth mentioning promising new technologies such as eDNA for auditing a wider variety of marine biodiversity?

Additional comments

It was interesting to read where the historical research has been focussed and where the research gaps are - this type of study could be very important for shaping the future of marine research in the region to ensure important knowledge gaps begin to be plugged.

I have one suggestion that I would highly encourage the authors to consider with a view to maximising the overall uptake of the finding of this study. While the reserach gaps are referred to in the discussion I would encourage the authors to go one step further and include an additional summary section/table along the lines of "Key Research Priorities to support marine biodiversity conservation in the RYS".

While this will no-doubt not be an exhaustive list - I think it would be really useful and make the document even more accessible if such a 'future research roadmap' could be created. The authors may want to consider including notes about the importance/significance/value of each research priority (or research question) and potentially rank each according to their perceived 'achievablity'. Within this table priority regions could be included (aka more reserach on all taxa is needed in the Tokara Group), or for example you could highlight more research on Echinodermata is needed in the Yakutane and Miyako Groups.

·

Basic reporting

Overall, I struggle to find the significance and relevance of the work as currently written in the paper. The authors need to do a better job of describing the significance of findings and placing results in the context of specific past work done. Replace general statements with some specific details, while also making the writing more concise.

Hypotheses are not clearly described and validated.

What are you trying to show with Table 1?- it is very difficult to read. Wikipedia is not an acceptable source of information-- do more research to locate acceptable references. Some of the information is relevant and should be referenced more in the paper, such as the relative sizes of the regions and population, but it currently seems buried in this difficult to read table. If you do not reference certain information in the paper, decide if it provides relevant information that would help explain research patterns, and use it in the discussion instead of burying it in a table. Much of the information would be much better presented as a figure. For example, you could show the marine research stations, labeled islands, land area and/or population size in a map relative to the number of papers or OBIS records found in these regions.

Your figures/table should clearly demonstrate the taxonomic and regional bias that you found, such that the reader does not have to search and struggle to find for the evidence of bias.

Overuse of terms like “urgent”, “stark”, “critical”, “clear” especially given the lack of clear patterns as currently presented in the text/figures.

Experimental design

Literature search methods are unclear, and methods section needs to be revised. Some more specific comments are in the attachment.

We need more information on relative rates of research… I am not convinced about the lack of research in the region because you need to back it up with relative figures (relative to land area, distance to research stations/university, population size, overall known biodiversity) and relevant background work. Figures that clearly present some of this information would help.

The analysis of OBIS records should not be described as an indicator of research rates, but instead as online accessible occurrence records. This is valuable in a different way, as these records can be easily downloaded and used for species distribution models, etc. In contrast, data from published research is often relatively inaccessible. As I mention in specific comments, OBIS records come from a variety of sources, not all a result of scientific research. I suggest including a figure with the number of OBIS records for one or more taxa that are particularly interesting in the paper, instead of supplement. Label the regions more clearly in the figure, and include other information that might help explain bias (i.e. location of research stations, populous cities, etc).

Validity of the findings

The discussion needs to be completely rewritten with more background information and context to concisely describe the significance of work in an organized and logical way.

Clearly lay out all of the potential sources of bias and how they relate to your results. One key driver of bias is likely the activity of one or more specialists in a region. You should look into whether certain specialists are driving any patterns found. For example, by looking at the collector/observers from OBIS records.

Why are the numbers of papers increasing?

After reading this paper, I also have very little idea of the content of work that has been published.

---

## Round 0.2 · accepted · Accept

Thank you for your detailed responses to the reviews. I am now happy to accept your interesting analysis for publication in PeerJ.

# ·

Basic reporting

The suggested improvements have been met - thank you.

Experimental design

The suggested improvements have been met - thank you.

Validity of the findings

The suggested improvements have been met - thank you.

Additional comments

Thank you for taking on board my comments and updating the manuscript.

·

Basic reporting

The current version of this article is much improved, and well written. I find that the authors sufficiently address comments from previous version.

Experimental design

The methods are more clearly explained and questions defined.

Validity of the findings

no comment